# An Alternative to Vermiculite: Composting on Tropical Islands Using Coral Sand to Enhance Nitrogen Retention during Ventilation

**Peng Cheng** [1,2], **Liqun Jiang** [3,*], **Rui Shan** [4,5], **Zhen Fang** [6], **Nianfang Ma** [3,*], **Lianwu Deng** [7,*], **Yaoquan Lu** [8], **Xiangping Tan** [2,5], **Weijun Shen** [2,5] **and Rongrong Liu** [1]

1   Institute of Environmental Research at Greater Bay Area, Key Laboratory for Water Quality and Conservation of the Pearl River Delta, Ministry of Education, Guangzhou University, Guangzhou 510006, China
2   Key Laboratory of Vegetation Restoration and Management of Degraded Ecosystems, South China Botanical Garden, Chinese Academy of Sciences, 723 Xingke Rd., Tianhe District, Guangzhou 510650, China
3   Guangdong Engineering Laboratory of Biomass High-Value Utilization, Guangdong Plant Fiber Comprehensive Utilization Engineering Technology Research and Development Center, Guangzhou Key Laboratory of Biomass Comprehensive Utilization, Institute of Biological and Medical Engineering, Guangdong Academy of Sciences, Guangzhou 510316, China
4   Guangzhou Institute of Energy Conversion, Chinese Academy of Sciences, Guangzhou 510640, China
5   Southern Marine Science and Engineering Guangdong Laboratory (Guangzhou), Guangzhou 511458, China
6   Biomass Group, College of Engineering, Nanjing Agricultural University, 40 Dianjiangtai Road, Nanjing 210031, China
7   Guangzhou Liushun Biotechnology Co., Ltd., Guangzhou 510030, China
8   Fujian Zhuangyuan Tea Co., Ltd., Sanming 354400, China
*   Correspondence: liqun_jiang2508@126.com (L.J.); nianfangma@hotmail.com (N.M.); dlwhhh@126.com (L.D.)

**Abstract:** Reducing nitrogen loss during composting with forced ventilation was comprehensively investigated in this study. Coral sand was tailored in the co-composting in the co-composting of sludge and litters. The physicochemical results revealed that forced ventilation prolonged the thermophilic phase and accelerated the substrate decomposition. With the addition of 10% native coral sand, the amount of nitrogen loss decreased by 9.2% compared with the original group. The microbial community evaluation revealed that the effect of forced ventilation on colony abundance was significantly greater than that of adding coral sand. This study demonstrated that when composting on a tropical island, adding coral sand under forced ventilation was a viable solution for realizing sustainable development.

**Keywords:** compost; coral sand; sludge; nitrogen emissions; forced ventilation

## 1. Introduction

Urbanization is spreading throughout the world, and subtropical islands are no exception. Disposing of municipal waste while maintaining a balance between ecological and urbanization is a crucial challenge to island environment [1,2]. Disposal of degradable components often faces with a choice of technical routes to meet various requirements under different scenarios. As islands are not as spacious or accessible as the mainland, there are fewer options for the in situ treatment of municipal waste on islands.

Composting is a biochemical process, where microorganisms degrade organic matters and finally generate stable humus [3]. The products can be used to improve soil fertility and realize waste utilization [4,5]. Municipal waste contains a high proportion of biodegradable substrates, and the compost products necessary feedstocks for the production of vegetables on islands [6]. Moreover, cost calculations have shown that the cost of composting was much lower than other treatment methods, such as trans-shipment, landfill, and incineration, especially for subtropical islands with strict environmental requirements for a minimal

impaction on the original ecology [7,8]. In summary, composting is a suitable solution for municipal waste treatment on islands.

However, the odor caused by forced ventilation was one of the major drawbacks of composting on an island. The main component of the odor is the ammonia volatilized from protein degradation [9,10]. Although forced ventilation improves the oxygen exchange conditions inside the composting heap, it releases a large amount of ammonia that can not only cause the loss of nitrogen but also affect the quality of life for island residents [11].

Recent studies [12,13] have explored several additives to reduce $NH_3$ emission by enhancing the microbial metabolism and facilitating the composting process, including struvite crystallization, zeolite, biochar, bentonite, vermiculite, ceramsite, and semi-coke [14–20]. The common characteristics of these additives include a large specific surface area, high ion exchange capacity, and relatively small particle size [21]. These characteristics can effectively enhance the oxygen exchange into the compost pile, and provide abundant pores for the ammonia adsorption to avoid volatilization during ventilation. Based on the above characteristics, a subtropical island native material-coral sand-was developed as an additive applied to the composting, which is economical accessible and could be exploited for the in situ composting.

In this work, natural coral sand was applied to compositing on a subtropical island. The physicochemical properties and microbial dynamic changes of the compost were comprehensively investigated to evaluate the impact of the additive on nitrogen loss, microbial metabolism, and capability of converting into fertilizer. To the best of our knowledge, this is the first study on composting additives guided by the in situ treatment principle by using native natural resource for soil amelioration.

## 2. Materials and Methods

### 2.1. Raw Materials and Pretreatment

The municipal waste was mixed with leaf litter to adjust the ratio of C/N and the initial moisture content. All raw materials were obtained from Yongxing, a subtropical island in the South China Sea. The materials were cut into small pieces of approximately 5 mm using an electrical grinder and mixed well before composting. The coral sand with a particle diameter of 5–10 mm was collected from the seashore of the island. The four experimental groups were labeled as control (C), ventilation (V), coral sand (S), and ventilation and coral sand (VS). In the forced ventilation groups (V and VS), the ventilation rate was 1 L/min (He et al., 2018) [18], and in the coral sand groups (S and VS), the amount of additive was 10% (proportions by weight). The properties of the raw materials are shown in Table 1.

**Table 1.** Properties of the substrates [1].

| Parameters | Raw Material | | |
| --- | --- | --- | --- |
| | Municipal Sludge (Pre-Dried) | Litter (*Pisonia grandis*) | Coral Sand (from Beach) |
| Total organic carbon, TC (%) | 67.51 ± 0.22 | 52.51 ± 0.17 | 0.18 ± 0.02 |
| Total nitrogen, TN (%) | 4.76 ± 0.05 | 1.73 ± 0.02 | 0.00 |
| C/N | 14.18 | 30.35 | - |
| pH | 6.91 ± 0.13 | 5.73 ± 0.09 | 7.31 ± 0.11 |
| Electrical conductivity, EC (μS/cm) | 505.64 ± 22.5 | 1102.41 ± 30.5 | 37.32 ± 0.06 |
| Moisture content (%) | 82.06 ± 0.47 | 61.43 ± 0.34 | 0.12 ± 0.02 |

[1] The values were means of triplicate experiments.

### 2.2. Composting Device Construction

The composting system employed in this study was cylindrical polyethylene reactors with dimensions of 320 mm × 600 mm × 6 mm (diameter × depth × thickness). The effective volume was 50 L. During the entire composting process with ventilation, air was pumped into the reactor using an aerator pump with a fixed flow rate of 1 L/min. Holes

were drilled on the lid of the reactor to place the thermometer probe and on the bottom to allow for leaching. To prevent heat loss, the reactors were wrapped with thermal-insulating material made of sponge and aluminum foil.

### 2.3. Analytical Methods and Calculation

The experimental samples from each of the four groups were divided into three parts: the first part was stored at −20 °C for the determination of microbial biomass; the second part was stored at −80 °C for the 16S rDNA and qPCR analysis, and the last part was used for the determination of moisture content, nutrient element content, and the evaluation of maturation after being dried at 65 °C. Physicochemical parameters (temperature, pH, and moisture content) were tested every day using thermometers (WSS301, DONTA, Shanghai, China), pH meter (PHS-3C, DONTA, Shanghai, China), and the thermogravimetric method (He et al., 2018) [18], respectively. Moreover, the below parameters were sampled on 0, 4, 7, 14, 21, 28, 35, 42, and 48 days. Total organic carbon (TC) and total nitrogen (TN) were determined using an elementary analyzer (Elemental Vario Micro, Elementar, Langenselbold, Germany) on dried and ground samples. Nitrate nitrogen was determined using an ultraviolet spectrophotometer from 5 g of fresh sample extracted with 100 mL KCl extraction solution. Ammonium nitrogen was determined by KCl extraction indophenol blue colorimetry. Electrical conductivity (EC) was measured using a conductometer (SX-650, DONTA, Shanghai, China) after 1:10 aqueous extraction ($w/v$, wet weight basis) of the fresh compost with deionized water. The germination index (GI) was determined as follows: distilled water was added to the compost sample in the ratio of 1:20, shaken at room temperature for 30 min, then boiled for 30 min and heated at 60 °C for 3 h. After filtration, 10 mL of the extract was put into a Petri dish covered with filter paper, and 100 Chinese cabbage seeds were sown in the dish. The Petri dishes were cultured at a constant temperature of 20 °C, and the index was calculated according to the number of germinated seeds.

DNA was extracted using a DNA Kit (Power soil DNA isolation kit, Mobio, Carlsbad, CA, USA). The total concentration of extracted DNA was determined by a spectrophotometer (Nanodrop 2000, Thermo Fisher Scientific Inc., Waltham, MA, USA). PCR products were recovered using a purification kit (Gene Jet, Thermo Scientific, Waltham, MA, USA). A Select Library Building Kit (NEB next) ® (Ultratm DNA library prep kit for Illumina, New England Biolabs, Ipswich, MA, USA) was used to construct the library. After the constructed library passed qubit quantitative and library detection, MiSeq was used for sequencing. Operational taxonomic unit (OTU) clustering and a species classification analysis were carried out based on effective data, and OTU and species annotations were combined to obtain the basic analysis results for OTUs and the taxonomic lineage of each sample. Then, the abundance and diversity indices of OTUs were calculated, and the species annotation was statistically analyzed at each classification level.

## 3. Results and Discussion

### 3.1. Variation of Physicochemical Properties during Decomposition

Temperature is one of the most important parameters during the mature process of compositing substrate. Generally, the compositing process could be divided into mesophilic phase, a thermophilic phase, and mature phase in terms of temperature change. In detail, the continuous thermophilic phase can effectively inactivate pathogenic bacteria and parasites, and promote the degradation of macromolecules such as cellulose and lignin. The temperature changes in the composting process of each group are shown in Figure 1a. Within 48 h after the initiation of composting, each experimental group was rapidly heated, and then entered the thermophilic period. The maximum temperatures of the V and VS groups were significantly higher than those of the C and S groups, indicating that the vigorous metabolism of microorganisms in the thermophilic phase could consume a large amount of oxygen. Meanwhile, the thermophilic durations of the ventilation groups (V and VS) were also longer than those of the other two groups, indicating that the macromolecular

organic substances such as cellulose and lignin in the substrate were not completely decomposed as compared to the effect of forced ventilation. Forced ventilation could enhance the activity of thermophilic microorganisms and increase the degradation efficiency in the thermophilic phase. As shown in Figure 1a, the thermophilic phase of various compost samples lasted for 7–9 days, which was a sufficient time to fully reach the hygienic standard. During the mature phase, due to the consumption of the substrate, the reactor could not maintain most heat, resulting in a significant temperature decrease. In the meantime, the thermophilic microorganisms became active again. The experimental cycle took place in late spring, and the room temperature was maintained at approximately 27 °C. The stack temperature of the V groups (V and VS) was slightly lower than that of the C group and S group, indicating that in the mature phase, the ambient ventilation could facilitate the oxygen consumption of the composting microorganisms, while the forced ventilation accelerated the heat loss and further reduced the stack temperature. If the ambient temperature was low in autumn and winter, then the ventilation might remove a large amount of heat during the cooling stage, thereby affecting the ripening efficiency. In each group, mesophilic phase was the composting cycle that was not affected by ventilation. At this phase, microorganisms adapted to the environment, and degradable organic matter such as saccharides and low-molecular-weight proteins were preferentially decomposed by these microorganisms. Due to the large number of miscellaneous bacteria in environmental samples, the demand for oxygen had not been selected by the composting process, and thus ventilation had little effect on the rapid temperature rise at this stage. Compared with the sufficient oxygen and heat dissipation brought by forced ventilation, an appropriate C\N ratio and moisture content had greater impacts on microbial metabolism in the mesophilic phase.

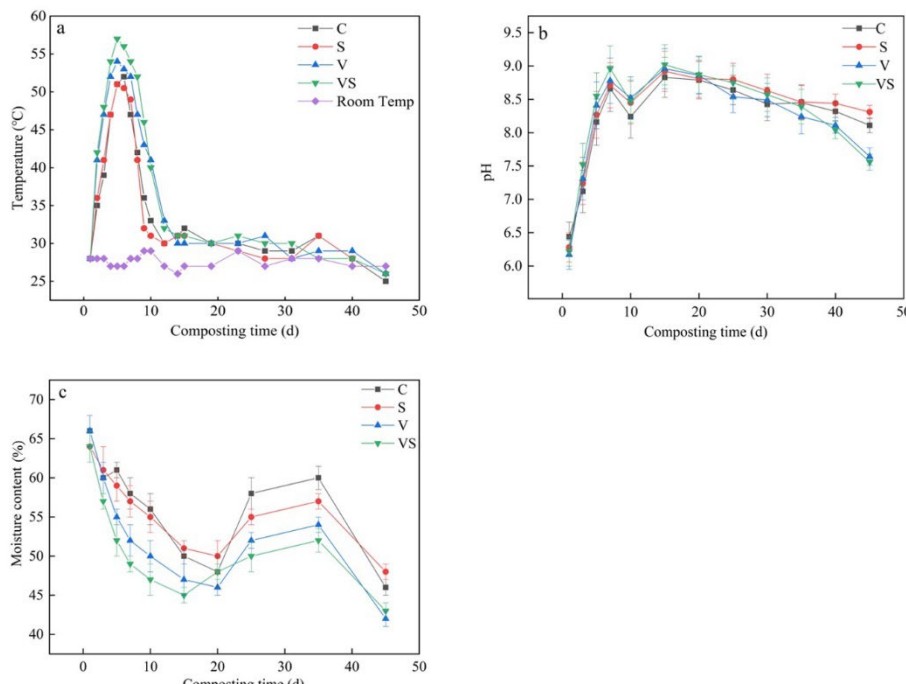

**Figure 1.** Changes in physicochemical properties during composting of different experiment groups (control (C), ventilation (V), coral sand (S), ventilation and coral sand (VS)): pile temperature and ambient temperature (**a**), pH (**b**), and moisture content (**c**).

The pH value was also an important parameter affecting the composting process. During the microbial degradation on the substrate, macromolecular organic matter was decomposed to produce ammonia, organic acids, and other products that could significantly affecte the pH value. Therefore, the pH value varied dynamically during the entire composting cycle. An appropriate pH could enable microorganisms to metabolize efficiently

and retain nutrients such as nitrogen in the substrate. As shown in Figure 1b, the initial pH of the reactor was low (6.0~6.5). The moisture content of municipal sludge was high during sludge discharge, close to an anaerobic state for microorganisms. Under this condition, organic matter was decomposed into organic acids, resulting in the weak acidity of the sample environment. When the reactor temperature gradually rose, degradable proteins in the mesophilic phase were metabolized by microorganisms to produce a large number of alkaline substances such as ammonia, causing a rapid rise in the pH. In the thermophilic phase coupled with ventilation, with the gas escaped, the pH slightly decreased in the middle of the thermophilic phase. After the temperature decreasing and the compost entering the mature phase, the pH of the reactor began to gradually decrease, from weakly alkaline to neutral. In the thermophilic phase, the main decomposed substrates were macro-molecules such as cellulose and lignin, and their products were saccharides. Therefore, with the escape of gas and the suspension of protein decomposition in the thermophilic phase, the number of miscellaneous microbial bacteria in the mature phase were greatly reduced. The above factors led to a small decline in pH and eventually to a constant level. In the last stage of the composting cycle, the pH values in the ventilation groups were further reduced to 7.55 (V) and 7.5 (VS) due to the large amount of gas exchange and additional nitrogen loss, which were similar to the values in neutral pH conditions. Although there was a certain loss of nutrients, the neutral composting products had little impact on the growth of crops and were not prone to adverse reactions such as root burning.

The change in moisture content in the composting process was derived from inter-action in the sample, the microbial metabolism, and the ventilation exchange. Moisture content has also been shown to correlate with the pile structure. As shown in Figure 1c, the moisture content of the whole compost sample showed periodic changes in different stages of composting. In the mesophilic phase of the experiment, the moisture content of the samples was greater than 65%. With the rapid increase in the reactor temperature, the moisture content decreased rapidly in the form of volatilization and metabolic consumption, and the decrease in moisture content was directly related to the ventilation effect. In the thermophilic phase, the moisture content of the C group was higher than that of the S group, followed by the V and VS groups, indicating that coral sand combined with forced ventilation could significantly enhance gas exchange during the thermophilic phase and increase microbial metabolism. The decomposition of organic macromolecules such as hemicellulose decomposed into small molecules could consume a large amount of water. There was a significant difference in moisture content among the forced ventilation groups (V and VS) and the other two groups in the thermophilic phase, up to 5–7%, indicating that the moisture loss at this stage was mainly due to the ventilation process.

After entering the mature phase, the temperature of the pile body decreased. At this time, the moisture content gradually increased, and a large amount of water vapor appeared on the cylinder wall of the composting equipment. Owing to the moisture released from the pile body without ventilation, the humidity of the C group and S group increased rapidly to the level of the original sample. During the mature phase, microorganisms continued to degrade the substrate and consume water, and the humidity of the pile began to decrease gradually. At the end of the mature phase, the moisture content of the V and VS groups had fallen to less than 45%; this met the basic requirements for biological fertilizer. The moisture content of the ventilation groups (V and VS) maintained a low level in the middle and late periods of the composting cycle; this was helpful for the circulation and exchange of gas in the reactor. However, water was an indispensable feedstock for microbial metabolism, and therefore a low moisture content could significantly affect the degradation of the substrate. The co-composting system of litter and municipal sludge refers to the substrates with high moisture content. During the mixing of composting raw materials, in order to make the microorganisms in the mesophilic phase quickly compost and metabolize, the C\N ratio was adjusted to 25–30. Due to the high moisture content of municipal sludge, the initial substrate moisture content of the final co-composting was also above 65%. Therefore, in this experiment, although forced ventilation would remove the moisture of the pile and

reduce the moisture content, each sample could still be ventilated, and coral sand could be added to enhance the gas exchange inside the reactor and improve the oxygen supply to microorganisms.

### 3.2. Changes in Organic Matter Loss with Coral Sand Addition

Total organic carbon (TC) is the total amount of organic matter represented by the content of carbon. As shown in Figure 2a, the TC of each experimental group showed a downward trend as a whole, and tended to be stable in the later stages. The TC decreased rapidly in the early stages, and then increased slightly after entering the thermophilic phase. In the mesophilic phase, the reactor was rich in degradable organic matters, and microbial metabolism was more active with abundant substrates. In the thermophilic phase, there were decompositions happening to macromolecular organic matters such as lignin and cellulose. The viable microbial species were relatively few, leading to the reduction of reduction rate. After the temperature dropped, the compost entered the mature phase. Although the temperature was again suitable for the metabolism of various microorganisms, the decomposition rate of the TC was still slow, due to the inactivation of a large number of miscellaneous bacteria in the thermophilic phase. After 45 days of composting treatment, the organic matter content of the C group and VS group decreased from 75.34% and 75.22% to 64.51% and 62.59%, respectively. The organic matter of the experimental group decreased significantly, and the metabolism of microorganisms was more vigorous, indicating that good ventilation was conducive to the degradation of substrate organic matter. Science it was necessary to add external bacteria to speed up the composting rate, it could be added in the mature phase to improve the ripening effect.

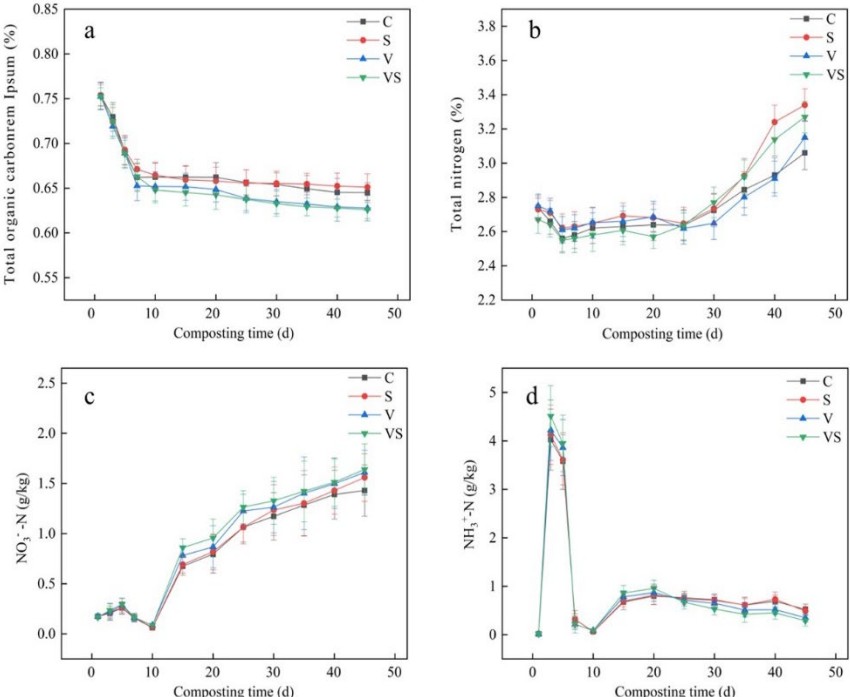

**Figure 2.** Changes in TC (**a**), TN (**b**), $NO_3^-$-N (**c**), and $NH_3^+$-N (**d**) during composting of different experiment groups (control (C), ventilation (V), coral sand (S), ventilation and coral sand (VS)).

The circulation of nitrogen has been validate with a crucial role in plant growth. Nitrogen in organic substrates could return to the earth in the form of ammonia nitrogen, and its transformation rate was related to microorganism nitrification and denitrification, and also oxygen content in the environment. In an aerobic environment, organic nitrogen would first be transformed into ammonia nitrogen, then nitrate nitrogen through nitrification. Ammonia nitrogen and nitrate nitrogen could be absorbed and utilized by plants, but ammonia nitrogen would be transformed into ammonia, finally escaping under alkaline

conditions. Under anoxic conditions, nitrate would also be converted into nitrogen by denitrification. Compared with the common nutrient elements P and K in other organic fertilizers, N was most likely to be gasified and escape. Therefore, coral sand and forced ventilation conditions would also have a significant impact on nitrogen circulation.

Composting was a reduction process where the organic matter was decomposed by microorganisms, finally converting these products to biological fertilizer. Overall, the TN of each experimental group showed an upward trend, indicating that the reduction process of organic matter was primarily through the reduction of organic carbon, resulting in the increase in the proportion of total nitrogen. In the mesophilic phase of composting, the TN decreased gradually (Figure 2b). This process was the decomposition of easily decomposed organic matter such as protein and lipid. During this process, a large amount of gas would be produced. The reduction rate of organic carbon was not as rapid as that of organic nitrogen, and thus the TN decreased. After entering the thermophilic phase, the decomposition of nitrogen-containing organic matter such as protein gradually stopped. In this process, the decomposition of organic macromolecules such as lignin and cellulose occurred, and the TC would gradually decrease. After the temperature of the reactor decreased and entered the mature phase, the temperature was again suitable for the metabolism of most microorganisms, and the reduction process of organic carbon accelerated. However, in this cycle, the proportion of nitrogen-containing organic matter was not high, so the conversion rate of organic nitrogen to inorganic nitrogen was relatively low. The TN gradually rose during the mature phase, finally reaching 3.06%, 3.34%, 3.15%, and 3.27% for the C, S, V, and VS groups, respectively. The addition of coral sand would improve the gas exchange inside the reactor, eliminating the anaerobic environment, and improve the fertilizing capacity of the products. Although ventilation would increase the oxygen supply and enhance microbial metabolism, it also took away a large amount of ammonia and reduced the fertilizing capacity. Further research is needed to explore the balance point between the quantitative treatment of organic matter reduction and composting.

As shown in Figure 2c, nitrate nitrogen generally emerged an upward trend and decreased significantly in the thermophilic phase, but the content increased rapidly after entering the mature phase. At the mesophilic phase of composting, nitrogen in organic matters decomposed into ammonium nitrogen, and then further transformed into nitrate nitrogen. After the temperature of the co-compost pile rose and the compost transited into the thermophilic phase, the transformation of organic nitrogen to ammonium nitrogen, then nitrate nitrogen would gradually shutoff; the microorganisms were deactivated or dormant at high temperature, and the nitrate nitrogen content reached the lowest value around the thermophilic phase on the 10th day of the composting cycle. When the temperature decreased, with the progress of composting, the microorganisms related to the nitrogen cycle gradually recovered; at the same time, the organic carbon in the substrate was gradually reduced and decomposed, and the proportion of nitrate nitrogen gradually increased in the composting products. In a closed composting system, due to the further decline in the moisture content of the pile in the later stages, the gas exchange of the pile, and the fast nitrification of microorganisms, the content of nitrate nitrogen in the mature phase was much higher than that in the initial period. In this study, they reached 1.43, 1.56, 1.61, and 1.64 g/kg for the C, S, V, and VS groups, respectively. Comparing the V group with the C group, ventilation was found to probable affect the total amount of nitrogen in the form of removing ammonia. However, for ionic and refractory nitrate nitrogen, it had only positive effects in increasing oxygen content and promoting microbial nitrification.

The conversion of nitrogen-containing organic matter such as protein to ammonium nitrogen was very rapid, which happened quickly in the mesophilic phase of composting. Figure 2d illustrated how the ammonium reached a maximum before entering the thermophilic phase, where the maximum ammonium nitrogen content of the VS group reached 4.51 g/kg during the cycle of composting. However, after entering the thermophilic phase, with the increase in pH and high temperature, the conversion of ammonium nitrogen to nitrate nitrogen gradually ceased; a large amount of ammonia was generated, and the

content of ammonium nitrogen decreased rapidly. During the mature phase, the content of nitrogen-containing organic matter decreased. The conversion of ammonium nitrogen to nitrate nitrogen continued to decline steadily, and finally maintained stable. At the end of the mature phase, the content of ammonium nitrogen decreased to 0.42, 0.39, 0.35, and 0.29 g/kg for the C, S, V, and VS groups, respectively. According to the bio-organic fertilizer standard of the China Ministry of Agriculture, when the ammonium nitrogen of the compost product was lower than 0.43 g/kg, it could be regarded as completely decomposed. Forced ventilation could significantly reduce the content of ammonium nitrogen in the thermophilic phase. A possible reason was that ventilation removed a large amount of ammonia and promoted the conversion of ammonium nitrogen. In the initial period, the influence of ventilation on the decomposition of organic matter such as protein was less effective than that in the mature phase. The reason for this might be that be the involvement of a transformation from organic nitrogen to ammonium nitrogen in initial period. In contrast, in the mature phase the transformation from ammonium nitrogen to nitrate nitrogen happened, and the metabolism of nitrifying bacteria was active in this process, causing a higher demand for oxygen. At the same time, at room temperature (27 °C), the pH of the reactor gradually decreased in the later stage, and the generation of ammonia was greatly restrained. Therefore, the loss of fertilizing ability caused by ammonium nitrogen ventilation was not as notable as the thermophilic phase.

### 3.3. Germination Index and Electroconductibility

When the seed germination index (GI) was above 50%, the material could meet the basic requirments of plant growth, while GI upper to 85% indicated that the fertilizer pile has been completely decomposed [22,23]. Figure 3a illustrated that during the mesophilic phase of composting, the GI of each group of samples was less than 50%, indicating that the samples could not yet meet the nutritional and sanitary needs of plant growth. The content of organic macromolecules in litter was high. Municipal sludge was rich in nutrients, but it could not be used without treatment, probable due to the presence of miscellaneous bacteria. With the composting process, the GI of each group of samples decreased significantly in the thermophilic phase, indicating that the composting products in the thermophilic phase could not meet the needs of crop growth. In the mesophilic phase, low-molecular-weight organic compounds were rapidly consumed via degradation by microorganisms. After entering the thermophilic phase, the dominant microorganisms in the fertilizer pile and their available substrate components became relatively fewer. In addition to meeting the sanitary conditions, seed germination also needed various nutrients and water. The samples in the thermophilic phase did not meet the above requirements. After the temperature of the reactor gradually decreased, thermophilic microorganisms began to reoccupy the dominant position in the reactor. Organic matter was degraded to form various nutrients, and the recovery of the moisture content of the reactor also stabilized the concentration of nutrients. In the V group, due to sufficient support and gas exchange, the substrate degradation was more efficient, as reflected in the change of GI. With the progress of composting, the germination index increased and gradually met the demand of plant growth. The addition of coral sand could accelerate the diffusion of gas in the reactor, but its low porosity made it impossible to alter the gas exchange efficiency when being added alone. Compared with the ventilation groups (V and VS), the addition of coral sand promoted the internal diffusion of the gas.

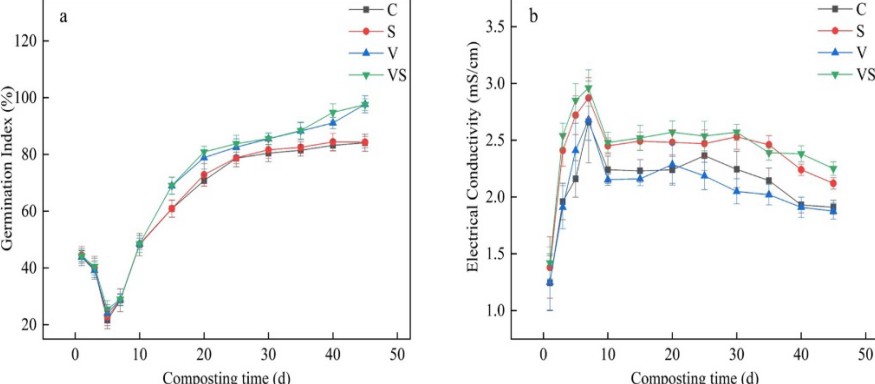

**Figure 3.** Changes in germination index (**a**) and electrical conductivity (**b**) of different experiment groups (control (C), ventilation (V), coral sand (S), ventilation and coral sand (VS)).

Electrical conductivity (EC) is a relevant parameter used to characterize the mineralization degree of the compost substrate detected by electrolytes. In general, a high the conductivity could demonstrate a high concentration of mineral ions, reflecting a greater impact on plants. An EC less than 4.0 MS/cm indicates that the concentration of electrolyte ions in compost products has no toxic effect on plants [24,25]. The overall change in the composting samples in Figure 3b showed a trend of rapid increase at the beginning and then gradually decrease. Due to the existence of various organic substances, the EC value in the initial composting samples was safe, but with the degradable organic matter decomposed by microorganisms in the mesophilic phase, the electrolyte ion concentration increased inevitably. The decomposition of organic matter might produce low-molecular-weight organic acids, and the main component of coral sand was calcium carbonate. Some organic acids such as carboxylate are more acidic than carbonate, resulting in the dissolution of a large amount of calcium ions. During the thermophilic phase, mineral salts began to form precipitates, and cations such as ammonium were reacted to produce ammonia volatilization. The EC value of the reactor began to decline again until the end of the mature phase. Although the EC value of the coral sand groups remained relatively high, values of 2.12 (S group) and 2.25 (VS group) could still meet the parameter standards of agricultural organic fertilizer.

### 3.4. Dynamic Changes in the Microbial Community during Composition

Most microorganisms in the environment are resistant of separated cultivation, and the culture temperature will also significantly affect the growth of different microorganisms [26]. Microorganisms with strong stress resistance will form spores and enter a dormant state. However, at low temperatures, most microorganisms will inactivate or simply enter a dormant state at high temperature. To quantitatively analyze the specific number of microorganisms, fluorescence quantitative PCR (qPCR) technology can be used to insert fluorescent genes into the PCR system. The change in the fluorescence signal during the entire PCR process is monitored in real time by a fluorescence-detection system, and finally compared with the standard curve for quantitative analysis. The overall change trend for the number of fragments of fungi was consistent with that of bacteria, but there were different changing trends in the mesophilic phase and mature phase. From the mesophilic phase to the thermophilic phase, the number of fungal fragments decreased to a certain percentage. For example, in the VS group, the number of fragments only decreased by 25%, indicating that the change in temperature in the ventilation groups (V and VS) had little impact on the number of fungi, and ventilation could effectively offset the impact caused by the temperature change. Among the known composting microorganisms, white rot fungi and brown rot fungi could tolerate high temperatures [27,28]. The cycle of substrate degradation required a thermophilic phase, but the temperature was low in the mesophilic phase and mature phase, where the thermophilic microorganisms could not be deactivated.

The decrease in fungal fragments from the mesophilic phase to the high-temperature stage was due to the consumption of substrate and the change in temperature that led to the inactivation of miscellaneous bacteria. During the high-temperature stage, thermophilic microorganisms further formed dominant colonies, and the number of fragments increased slowly during the cooling stage.

It could be seen from Figure 4 that the overall change trend of the total amount of bacteria in the compost pile was opposite to the change of temperature. The maximum copies of bacterial 16S rDNA fragments were obtained in the mesophilic phase. This result was consistent with the high-throughput sequencing analysis data. The mesophilic phase was rich in organic substrate and the degradable organic matter could be utilized by various microorganisms. Meanwhile, the temperature was appropriate for metabolism in the mesophilic phase, so the miscellaneous bacteria could also grow and breed. Therefore, the sanitary conditions and safety of the composting products could not meet the standard. Due to the effect of high temperature, the copy of bacterial gene fragments significantly decreased in the thermophilic phase. For example, in the VS group, the number of fragments decreased by almost 82%. Most bacteria could not resist high temperatures, and only some bacteria with strong stress resistance could form spores and enter the dormant state during thermophilic phase. Comparing each experimental groups, a higher the temperature corresponded with a lower the copy of bacterial fragments in the thermophilic phase, indicating a more preferable sanitary condition of the fertilizer pile. During the mature phase, the number of bacterial fragments recovered to a certain extent, but the overall number was lower than that of the initiation. During the mature phase, the nutrients were relatively sparse, but the temperature was appropriate and there was no interference of miscellaneous bacteria. The surviving microorganisms would grow and reproduce quickly, to further accelerate the maturation of the fertilizer pile and improve the fertilizer.

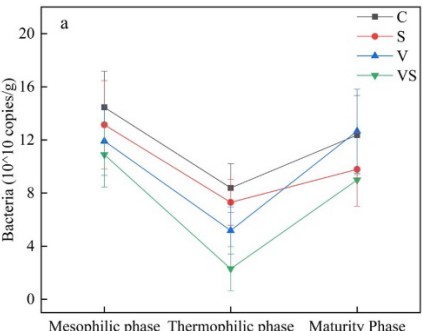 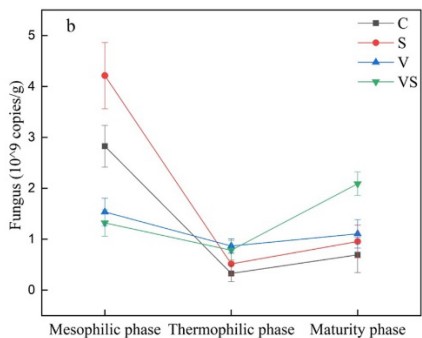

**Figure 4.** Dynamic changes in biomass (bacteria (**a**) and fungus (**b**)) during composting with qPCR analysis of different experiment groups (control (C), ventilation (V), coral sand (S), ventilation and coral sand (VS)).

The Chao index is often used to reflect community richness. A larger Chao index reflect a higher the microbial community richness. Besides, the Shannon index reflects the diversity of the microbial community, and larger values could indicate a more diverse bacterial community. Simpson's index, also known as the diversity index, is often used to reflect the diversity in the community. The larger the value, the lower the community diversity is. The variation trends of bacterial diversity and richness were basically the same, showing a trend of incipient decreasing, then slightly increasing, and finally tending to be stable. In the thermophilic phase of different experimental groups, the Chao index was 2292.98, 2277.82, 2180.08, and 2067.64 for the C, S, V, and VS groups, respectively. At the end of the thermophilic phase, the Chao index increased slightly. The dominant microorganisms in the thermophilic phase were thermophilic microorganisms, most of which were fungi, and thus led to the decreasing of bacteria Chao index. After entering the mature period, the temperature decreased and returned to the optimal range for mesophilic bacterial growth. Microorganisms multiplied in large numbers, thereby increasing the Chao index. The Shannon index of the compost samples in the heating period was less

than 5, while in the initial period and mature phase, the Shannon index was relatively high. In the mesophilic phase, due to the appropriate temperature and sufficient nutrition, the composting condition was conducive to the growth and breeding of bacteria, so the Shannon index was high, reaching more than 5. After entering the thermophilic phase, the environmental temperature was not conducive to the metabolism of bacteria, and the bacteria with strong stress resistance formed spores into dormancy, and the Shannon index decreased. Due to the consumption of substrate and the rapid heat dissipation caused by the large temperature difference between surroundings and the composting piles, the reactors entered the mature phase after the degradation of macromolecules such as cellulose. In this phase, the fungi withdrew from the dominant position; thermophilic microorganisms became active again, and the Shannon index gradually increased, although being less than the initial value in the diversity index table. The Shannon index in the mesophilic phase was 5.62, 5.33, 5.33, and 5.31 for the C, S, V, and VS groups, respectively, and were 5.31, 5.16, 5.12, and 4.91 in the mature phase. The reason might be that compared to the thermophilic phase, the nutrition was relatively rich in the mesophilic phase. The existence of various forms of organic matter led to the breeding of a large number of miscellaneous bacteria, and the nutritional composition was relatively sparse in the mature phase. Meanwhile, miscellaneous bacteria did not easily survive, and some bacteria with heat resistance and poor stress resistance were killed in the thermophilic phase. Simpson's index further verified the change trend of the Shannon index. The low Simpson's index in the mesophilic phase indicated that there were abundant microbial species, and the index in the thermophilic phase increased exponentially, indicating that the diversity of the bacterial community was greatly declined. After entering the mature phase, the index decreased rapidly, indicating that the diversity of the community was improved after cooling, thus enriching the species of microorganisms.

The alpha diversity indexes of fungi are shown in Table 2. The coverage index of each group was more than 0.99, indicating that the sequence detection probability of fungi in composting samples was high, and the sequencing results could effectively benchmark the real situation of the fungal community during composting. The change trends of the Chao index and sequence number of fungal samples were basically the same, showing a significant decline from the initial period to the thermophilic phase, but a slow decline from the thermophilic phase to the mature phase, indicating that there was natural selection of fungi from the initial period to the thermophilic phase, and the microorganisms adapted to the high temperature survived and bred, while other miscellaneous bacteria were deactivated. When the temperature drop is unsuitable for the metabolism of elevated temperature resisted fungi such as white rot fungi and brown rot fungi, the fungi will enter dormancy. Simpson's index reflects the diversity of the microbial community from the dimension of probability. In the mesophilic phase, the Simpson indices of samples in each experimental group were 0.06, 0.08, and 0.13 for the C, S, and V groups, while the Simpson index in the thermophilic and mature phases reached the maximum and the minimum 0.47 and 0.24, indicating that the diversity of the fungal community significantly decreased after the thermophilic phase, while the Simpson index in the thermophilic phase of the V and VS groups was as high as 0.47 and 0.42, indicating that ventilation effectively promoted the metabolism of thermophilic microorganisms at high temperature and greatly reduced the number of miscellaneous bacteria. The final response trend of the Shannon index was consistent with other parameters in the alpha diversity index table, showing a change trend of initially decreasing and then slightly increasing, indicating that the diversity of the fungal community decreased at first, and then slowly increased during the composting process.

**Table 2.** Community diversity index of bacteria and fungus during different phases of composting.

| Composting Stage | Sample (Bacteria) | Shannon | Simpson | Chao | Coverage | Sample (Fungus) | Shannon | Simpson | Chao | Coverage |
|---|---|---|---|---|---|---|---|---|---|---|
| Mesophilic phase (I) | CI | 5.62 | 0.02 | 2719.58 | 0.99 | CI | 4.03 | 0.06 | 1191.95 | 0.99 |
| | SI | 5.33 | 0.02 | 2628.01 | 0.99 | SI | 3.91 | 0.08 | 1117.12 | 0.99 |
| | VI | 5.33 | 0.03 | 2939.76 | 0.98 | VI | 3.57 | 0.13 | 1090.87 | 0.99 |
| | VSI | 5.31 | 0.06 | 2639.76 | 0.98 | VSI | 3.25 | 0.22 | 1042.44 | 0.99 |
| Thermophilic phase (R) | CR | 4.81 | 0.06 | 2292.98 | 0.98 | CR | 2.52 | 0.34 | 859.88 | 0.99 |
| | SR | 4.69 | 0.06 | 2277.82 | 0.98 | SR | 2.32 | 0.39 | 812.46 | 0.99 |
| | VR | 4.68 | 0.05 | 2180.08 | 0.98 | VR | 1.97 | 0.47 | 726.94 | 0.99 |
| | VSR | 4.19 | 0.07 | 2067.64 | 0.98 | VSR | 2.11 | 0.42 | 681.11 | 0.99 |
| Mature phase (D) | CD | 5.31 | 0.03 | 2517.76 | 0.98 | CD | 3.06 | 0.24 | 1022.02 | 0.99 |
| | SD | 5.16 | 0.05 | 2549.44 | 0.98 | SD | 2.73 | 0.31 | 967.16 | 0.99 |
| | VD | 5.12 | 0.04 | 2484.95 | 0.98 | VD | 2.64 | 0.33 | 848.34 | 0.99 |
| | VSD | 4.91 | 0.04 | 2442.79 | 0.98 | VSD | 2.72 | 0.32 | 898.81 | 0.99 |

At the genus level, the bacteria with the highest abundance were *Glutamicibacter*, *Pantoea*, and *Pseudomonas* (Figure 5). *Pantoea* could utilize various saccharides, which was a facultative anaerobic chemotrophic bacterium with strong stress resistance [29]. It could be seen that the relative content of *Pantoea* in each group of experimental samples decreased during the thermophilic phase, but increased significantly after entering the mature phase, indicating that *Pantoea* could become dormant during the thermophilic phase, and finally recovered its activity after cooling. Both *Glutamicibacter* and *Pseudomonas* could effectively utilize protein, but the utilization of saccharides was poor, being complementary to *Pantoea* in terms of raw materials [30,31]. The relative contents of *Glutamicibacter* and *Pseudomonas* were remarkably reduced in the thermophilic phase, indicating that the degradation of substrate by microorganisms in the thermophilic phase was based on components, such as cellulose and lignin, rather than low-molecular-weight saccharides and proteins. In addition, *Enterobacter*, *Curtobacterium*, and *Pseudomonas* could synthesize antibiotics to inhibit the growth of other microorganisms and reduce the content of miscellaneous bacteria [32,33]. In the experimental group, in each mature phase, the abundance of detected bacteria was significantly less than that in the initial period.

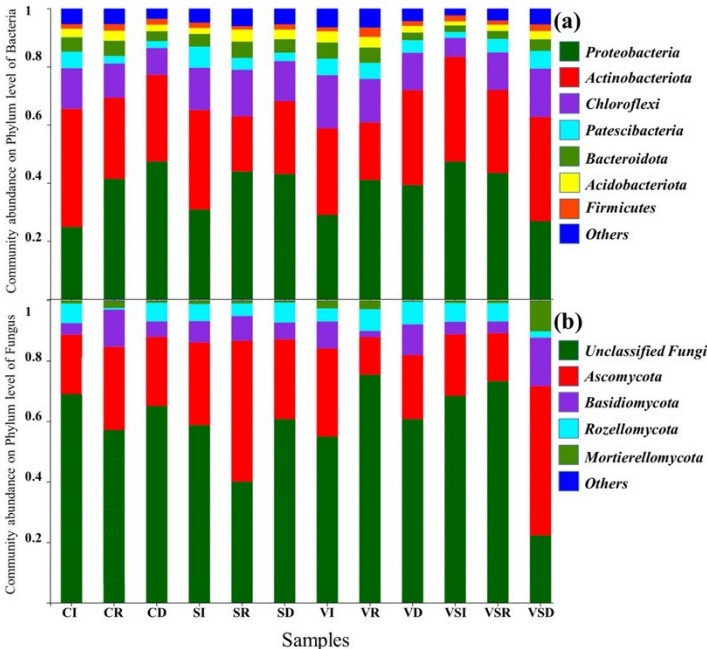

**Figure 5.** Community abundance at the phylum level of bacteria (**a**) and fungus (**b**).

As concluded in Figure 5, at the genus level, the thermophilic phase was significantly distinguished from the initial period and mature phase. The detection rate of fungi in

the thermophilic phase was greater than other phases, and the species richness was high, indicating that in the mesophilic phase, there were also thermophilic microorganisms existing in the substrate, but they were dormant. However, in the thermophilic phase, the species richness was greatly improved, indicating that high temperature could promote the metabolism of most composting fungi. After the reactor entered the mature phase and returned to ambient temperature, the richness of fungal colonies decreased rapidly, indicating that thermophilic microorganisms entered dormancy again at the final stage.

Co-composting has been widely used to adjust the initial C/N ratio. As a feedstock that was rich in nitrogen, the municipal waste of mature and sludge was mixed with litter to reduce the moisture content while promoting the C/N ratio to a proper range. During composting, the total nitrogen loss was mainly derived from the emission of $NH_3$, which did cause not only odor problem and global warming, but also reduce the quality of copost by-products. Additives were effective by adsorbing $NH_3$, reducing the pH of the pile and forming struvite. However, the special feature of the island depended on a balance needs to be achieved between the transportation cost and the improvement effect. As for a natural additive without modification, the $NH_3$ emission reduction from Zeolite [34], Bentonite [35], Ceramsite [36], Phosphogypsum [37], Superphosphate [38] was 18%, 3.39%, 21%, 21.65% and 18.9%, respectively. However, the application of coral sand on the islands was cost-effective and local-sustainable. To solve, bacterial power cultivated from the compost pile could be a considerable method for a further improvement. The addition of biochar in the co-composting process of mixing pig manure with sawdust resulted in a 13% $NH_3$ retention. When the additive was mixed with bacterial powders from the formal pile, the retention was promoted to 26% [39]. Another way was to adjust the ratio of additives. In the experiment with additives of biochar and microbial consortium, when the initial ratio was 1:5, the $NH_3$ retention was 21.8% [40]. The conservation rose to 41.4% when the ratio became 1:1, which revealed that even with exactly the same additive, there were still promotions achieved by varying the component ratio.

## 4. Conclusions

Addition of coral sand to reduce the nitrogen loss during the forced ventilation composting was comprehensively demonstrated in this study. Coral sand, as a raw material that could be readily obtained on a subtropical island, was used to substitute the vermiculite as a compost filler, leading to a decreased nitrogen loss by 9.2%. In addition, the moisture content of the pile decreased from 46% to 42%, which has been validated as a factor that was indicatable in the preservation and transportation of the mature compost pile. The results showed that compared with forced ventilation, adding coral sand did not affect the community of composting microorganisms, which effectively protected the local microorganisms and plant resources. The in situ disposal plan was feasible for other islands. Additionally, compared with other additives, coral sand was cost-effective, while the biological fertilizer produced by compost could be marketed on the islands, indicating that this study could potentially facilitate high economic benefits during ventilation compositing in the future.

**Author Contributions:** Conceptualization, P.C. and R.S.; methodology, P.C. and N.M.; validation, P.C. and X.T.; formal analysis, P.C. and Y.L.; investigation, P.C. and W.S.; data curation, P.C. and L.D.; writing-original draft preparation, P.C. and Z.F.; writing-review and editing, L.D. and L.J.; visualization, L.J. and N.M.; supervision, L.J. and Z.F.; project administration, R.L.; funding acquisition, P.C. and R.L. All authors have read and agreed to the published version of the manuscript.

**Funding:** This research was financially supported by Postdoctoral Science Foundation, China (2020M672866), National Natural Science Funds of China (22106163), GDAS' Project of Science and Technology Development (2022GDASZH-2022010110), Key Special Project for Introduced Talents Team of Southern Marine Science and Engineering Guangdong Laboratory (Guangzhou, GML2019ZD0101).

**Institutional Review Board Statement:** Not applicable.

**Informed Consent Statement:** Not applicable.

**Data Availability Statement:** Not applicable.

**Conflicts of Interest:** The authors declare no conflict of interest.

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
