# Peer review of "An Alternative to Vermiculite: Composting on Tropical Islands Using Coral Sand to Enhance Nitrogen Retention during Ventilation"

_fermentation, doi:10.3390/fermentation8100552_

Round 1

Reviewer 1 Report

This article examines the usage of coral sand in composting for the prevention of nitrogen release to the environment. The scope of the paper is to provide new knowledge for understanding the role of coral sand to the compost studying the physicochemical properties and the microbial dynamic changes of it. 

The paper certainly meets the aim and the scope, as well as the high academic standards of the ‘Fermentation’ Journal. However, the following specific improvements should be made, before accepting the paper for possible publication to the Journal.

·       - In the begging of the Introduction, more information is suggested to be added for what part of the municipal wastes is used for the compost process.

·        Line 43-46, the contradiction here confuses the reader if it the best solution or not to compost on islands. Can you please revise it?

·      -  In line 60-61, proposing coral sand as free of charge, free to all people for aiding composting. However, is it sure that no permission needed for the exploitation of this native material of the island?

·    -    Results and Discussion session is too long. It is recommended to be reduced in size.

·    -    It is proposed to add more references comparing your results with other studies.

·    -    Conclusions are poor considering the long part of the results. You have to revise it with additional information obtained from your results. It is recommended to discuss the marketability of the produced compost and the benefits to other islands.

Author Response

This article examines the usage of coral sand in composting for the prevention of nitrogen release to the environment. The scope of the paper is to provide new knowledge for understanding the role of coral sand to the compost studying the physicochemical properties and the microbial dynamic changes of it.

The paper certainly meets the aim and the scope, as well as the high academic standards of the ‘Fermentation’ Journal. However, the following specific improvements should be made, before accepting the paper for possible publication to the Journal.

Issue 1: In the begging of the Introduction, more information is suggested to be added for what part of the municipal wastes is used for the compost process.

Answer: Thanks for your kind suggestion, more information is added into the manuscript and the details are as follows:

Urbanization is spreading throughout the world, and subtropical islands are no exception. The problem of disposing of municipal waste while maintaining an ecological balance accompanies urbanization. It is effective to classify municipal waste, but the disposal on the degradable components often faced with the choice of technical routes to meet the needs of different scenarios. As islands are not as spacious or as accessible as the mainland, there are fewer options for in-situ treatment of municipal waste on islands.

Issue 2: Line 43-46, the contradiction here confuses the reader if it the best solution or not to compost on islands. Can you please revise it?

Answer: Yongxing Island is a small urbanized city with pleasant natural environment and few residents. Composting is obviously the best solution for municipal waste disposal to protect the original ecology. However, with the progress of society, efficiency became the first factor, such as pyrolysis and landfill. It is also our duty to promote biotreatment effect to achieve a harmonious unity of efficiency and environmental protection.

Issue 3: In line 60-61, proposing coral sand as free of charge, free to all people for aiding composting. However, is it sure that no permission needed for the exploitation of this native material of the island?

Answer: Coral sand is common natural material on the island, the ownership belongs to the local government, as well as the municipal facilities. According to our on-the-spot investigation, relevant local departments welcome us to use local materials to solve the composting problems.

Issue 4: Results and Discussion session is too long. It is recommended to be reduced in size.

Answer: Upon your suggestion, the Results and Discussion session is reduced for legibility.

Issue 5: It is proposed to add more references comparing your results with other studies.

Answer: Upon your suggestion, more references are added in the revised manuscript to compare the results and innovations.

Issue 6: Conclusions are poor considering the long part of the results. You have to revise it with additional information obtained from your results. It is recommended to discuss the marketability of the produced compost and the benefits to other islands.

Answer: Thanks for your kind suggestions, the conclusions are revised to highlight the marketability and the eco-benefits. The revised conclusions are as follows:

The feasibility of adding coral sand to reduce the nitrogen loss during forced ventilation composting was demonstrated in this study. Coral sand is a raw material that can be obtained everywhere on a subtropical island at almost no cost, and substituting coral sand for vermiculite as a compost filler decreased nitrogen loss by 9.2%. In addition, the moisture content of the pile decreased from 46% to 42%, a factor that was instrumental in the preservation and transportation of the mature compost pile. As coral sand is a local material, the experimental results showed that compared with forced ventilation, adding coral sand did not affect the community of composting microorganisms, which effectively protects the local microorganisms and plant resources. The in-situ disposal plan is feasible for other islands. Additionally, compared with other additives, coral sand has almost no cost, while the biological fertilizer produced by compost can be marketed on the islands, indicating that this study has high economic benefits.

Reviewer 2 Report

Dear Authors, 

Thank you for submitting your manuscript to Fermentation! The manuscript provides an interesting study with a scientific methodology. There are instances where the manuscript requires further improvement. 
*Line 35 How does the land usage in composing is less than the in-situ municipal waste treatment on islands?
*Line 41 What are high environmental requirements? The manuscript should be self explanatory to the reader and should give them all the information needed to understand the manuscript. 
*Line 43 the argument "composting is the suitable solution for municipal 43 waste treatment on islands." is not well supported. 
*Line 53-54 Effective of/on what?
*The abbreviations (TC, TN etc.) in the table 1 do not come at the time when the table is mentioned in the text. The abbreviation need to first mentioned against first use. 
*How was the size of the coral sand determined? 
*Line 69 Adjustment of the C/N with respect to what? The C/N appears different in the table. 
*The properties of the substrates are means of triplicate experiments or measurements? If from experiment which experiment?
*Line 86, which lid?
*Why did the authors chose insulating material made of sponge and aluminum foil and not other material viz. glass wool commonly used for purpose? How good the temperature control was?
*Line 97: Other parameters were sampled? Which parameters?
*Line 128: How the continuous thermophilic phase can promote the hydrolysis of cellulose and lignin?
*Line 137, 141, : How did the authors made the claim? Is there any data to support it?
*Line 145: Will the results vary with different season other than late spring? 
*Line 151 if the ambient temperature is maintained at 27C why the results will be different in seasons other than late spring?
*Line 163: How is the claim supported by data or reference? Did the authors see any compounds produced that changed the pH?
*Line 171 What is weak acidity?
*The resolution of the graphs needed to be improved. It will be better if the graphs are place from left to right than from top to bottom, as the later pattern is easier to comprehend for the readers. 
* The results have been discussed adeptly but not compared to the previous work done. The discussion of the results need to be improved. 
* Sometimes the authors are explaining results in previous sections that is making it difficult to comprehend the data and explanation. 
* Line 379: Do all the microorganisms under stress conditions make spores?
*Line 416 How does the decrease in fungal fragments corelates with the inactivation of miscellaneous bacteria?
*The names of the organisms should be italicized in the text and the references. 

Thanks

Author Response

  • Reviewer #2

Thank you for submitting your manuscript to Fermentation! The manuscript provides an interesting study with a scientific methodology. There are instances where the manuscript requires further improvement. 
Issue 1: Line 35 How does the land usage in composing is less than the in-situ municipal waste treatment on islands?

Answer: At present, there is a composting workshop on Yongxing Island, and compost products will be used for planting vegetation on the island. Therefore, compared with landfilling, composting is a process of waste utilization which can  realize material recycling and reducing land usage.

Issue 2:
Line 41 What are high environmental requirements? The manuscript should be self-explanatory to the reader and should give them all the information needed to understand the manuscript. 

Answer: The high environmental requirements mean minimal impaction on the original ecology. Upon your suggestion, more explanations are added in the revised manuscript for a better understanding.

Issue 3: Line 43 the argument "composting is the suitable solution for municipal 43 waste treatment on islands." is not well supported. 

Answer: The reason why “composting is the suitable solution for municipal waste treatment on islands” are listed as below:

  1. The ecosystem of Yongxing Island is very fragile, most of the vegetation comes from artificial planting, which consumes plenty of fertilizer. Composting can produce fertilizer while disposing municipal organic waste, it is an eco-friendly project with good economic benefits;
  2. Landfill and incineration are common technologies for municipal waste treatment, but landfill will cause groundwater pollution and occupy valuable land resources, which is intolerable for islands with limited area. Incineration is efficient and occupied limited area, but waste gas will cause air pollution with wider range than odor from composting. Besides, the incineration device is expensive to transport from the mainland and needs special maintenance.
  3. To sum up the above arguments, although composting on the subtropical island is facing posers like odor and low maturity , the problems are solvable. Our on-the-spot investigation also found that the degradable municipal waste on Yongxing Island were composted in situ, while the non-degradable ones were packaged and shipped to the mainland for treatment.

Issue 4: Line 53-54 Effective of/on what?

Answer: In the revised manuscript, the express is “Struvite crystallization, zeolite, biochar, bentonite, vermiculite, ceramsite, and semi-coke have been shown to be effective in reducing NH3 emissions when added into the composting heap”, which corresponded to the above sentence: “Recent studies have examined the use of additives to reduce the NH3 emissions through enhancing the microbial metabolism and facilitating the composting process.”

Issue 5: The abbreviations (TC, TN etc.) in the table 1 do not come at the time when the table is mentioned in the text. The abbreviation need to first mentioned against first use. 

Answer: Thanks for your kind suggestion, the abbreviations are mentioned at the first appearance.

Issue 6: How was the size of the coral sand determined? 

Answer: The size of coral sand is set with reference to vermiculite. Your kind suggestion will lead us to consider the impact of additive size on composting in subsequent studies.

Issue 7: Line 69 Adjustment of the C/N with respect to what? The C/N appears different in the table. 

Answer: The initial C/N ratio of municipal sludge and litter is different, both of which are not suitable for compost. The purpose of co-composting is to adjust the initial C/N ratio to get a quick start-up. Recent studies revealed that a proper initial C/N ratio was 25~30 [1, 2].

Issue 8: The properties of the substrates are means of triplicate experiments or measurements? If from experiment which experiment?

Answer: In order to minimize the error, all experimental groups have set up three groups in parallel. At the same time, the three-point method is used to measure the average value.

Issue 9: Line 86, which lid?

Answer: In Line 86, “Holes were drilled on the lid to place the thermometer probe and on the bottom to allow for leaching”, the lid is on the top of the reactor, a cover for the can.

Issue 10: Why did the authors chose insulating material made of sponge and aluminum foil and not other material viz. glass wool commonly used for purpose? How good the temperature control was?

Answer: The selection of thermal insulation materials was referred to He et al [3]. Indeed, the glass wool recommended by the reviewer should be a better choice. Due to the full coverage of sponge and aluminum foil, the temperature was well controlled. The maximum internal and external temperature difference could reach 30℃.

Issue 11: Line 97: Other parameters were sampled? Which parameters?

Answer: In line 97: “Moreover, other parameters were sampled on day 0, 4, 7, 14, 21, 28, 35, 42, and 48”. The parameters are total organic carbon (TC), total nitrogen (TN), nitrate nitrogen, ammonium nitrogen, electrical conductivity (EC), germination index (GI). The above parameters appear in the following text: “Total organic carbon (TC) and total nitrogen (TN) were determined using an elementary analyzer (Elemental Vario Micro, Germany) on dried and ground samples. Nitrate nitrogen was determined using an ultraviolet spectrophotometer from 5 g of fresh sample extracted with 100 ml KCl extraction solution. Ammonium nitrogen was determined by KCl extraction indophenol blue colorimetry. Electrical conductivity (EC) was measured using a conductometer (SX-650, China) after 1:10 aqueous extraction (w/v, wet weight basis) of the fresh compost with deionized water. The germination index (GI) was determined as follows: distilled water was added to the compost in the ratio of 1:20, shaken at room temperature for 30 min, then boiled for 30 min and heated at 60 °C for 3 h. After filtration, 10 ml of the extract was put it into a Petri dish covered with filter paper, and 100 Chinese cabbage seeds were sown in the dish. The Petri dishes were cultured at a constant temperature of 20 °C, and the index was calculated according to the number of germinated seeds.”

Issue 12: Line 128: How the continuous thermophilic phase can promote the hydrolysis of cellulose and lignin?

Answer: The optimum temperature range for enzymes to hydrolysis refractory cellulose and lignin is 55°C ~ 65°C [4]. The continuous thermal phase can make the  relevant enzymes in the best activity so as to effectively degrade the substrate and achieve maturity.

Issue 13: Line 137, 141, : How did the authors made the claim? Is there any data to support it?

which was not conducive to the degradation of cellulose and lignin, as well as the sterilization of miscellaneous bacteria. Answer: According to the China's hygienic standard for the harmless treatment of feces (GB 7959-87), the temperature of compost products in the thermophilic phase must be above 55°C for five days to meet the sterilization requirements. In the experiment groups without forced ventilation. The maximum temperature and duration of the thermophilic phase were lower than those of the ventilation group,

Issue 14: Line 145: Will the results vary with different season other than late spring? 

Answer: In this study, the ambient temperature was controlled to study the impact of coral sand on the co-compost. The results vary from the two factors below: coral sand and forced ventilation.

Issue 15: Line 151 if the ambient temperature is maintained at 27C why the results will be different in seasons other than late spring?

Answer: The experimental materials came from Yongxing Island, a subtropical island with an annual average temperature of 23~29℃. In line 152: “If the ambient temperature is low in autumn and winter, the ventilation may remove a large amount of heat during the cooling stage, thereby affecting the ripening efficiency”. We mean that if the ambient temperature is not maintained, the experimental results would be affected by the ambient temperature in different seasons, which did not conform to the relatively constant temperature on the island throughout the year. The results difference came from the addition of coral sand and forced ventilation.

Issue 16: Line 163: How is the claim supported by data or reference? Did the authors see any compounds produced that changed the pH?

Answer: Composting is actually the process of microbial decomposition on organic matters, and the acidity of the initial stack comes from organic acid [5]. With the degradation of organic matter, such as the conversion of sugars and lipids to carbon dioxide, and the conversion of proteins to ammonia [6, 7], the pH of the co-composting piles increases. The dynamic changes of compounds in this process were inferred from the relative content change of total oxygen carbon, total nitrogen, ammonium nitrogen and nitrate nitrogen.

Issue 17: Line 171 What is weak acidity?

Answer: In Line 171: “the initial pH of the reactor was low and weakly acidic”. It was a syntax error and we have revised it as “weak acidic”

Issue 18: The resolution of the graphs needed to be improved. It will be better if the graphs are place from left to right than from top to bottom, as the later pattern is easier to comprehend for the readers. 

Answer: Upon your suggestion, the resolution and placement of the graphs are adjusted for easier reading. Our original intention is to share an abscissa for vertical comparison of data.

Issue 19: The results have been discussed adeptly but not compared to the previous work done. The discussion of the results need to be improved. 

Answer: Upon your suggestion, the discussion part is revised and the details are as follows:

Co-composting is a common method to adjust the initial C/N ratio. As a feedstock that is rich in nitrogen, the municipal waste of mature and sludge were mixed with litter to reduce the moisture content while promoting the C/N ratio to a proper range. During composting, the total nitrogen loss was mainly through the emission of NH3, which did not only cause odor problem and global warming but also the reduction of compost by-product quality. Additives are effective by adsorbing NH3, reducing the pH of the pile and forming struvite. However, the special feature of the island is that a balance needs to be achieved between the transportation cost and the improvement effect. As for natural additive without modification, the NH3 emission reduction from Zeolite, Bentonite, Ceramsite, Phosphogypsum, Superphosphate was 18%, 3.39%, 21%, 21.65% and 18.9%, respectively. From the perspective of the nitrogen retention, the effect of coral sand is average. However, the application of coral sand on the islands is cost effective and local resources protective, there it still room for improvement in additive performance. Bacterial power cultivated from the compost pile can be a considerable method. In the pig manure mixed with sawdust, the co-composting process added biochar, which resulted in a 13% NH3 retention. When the additive was mixed with bacterial powders from the formal pile, the retention was promoted to 26%. Another feasible practice is to adjust the ratio of additives. In the experiment with additive of biochar and microbial consortium, when the initial ratio was 1:5, the NH3 retention was 21.8%. The conservation rose to 41.4% when the ratio became 1:1, which revealed that even with exactly the same additive, there is still promotions achieved from varying the component ratio.

Issue 20: Sometimes the authors are explaining results in previous sections that is making it difficult to comprehend the data and explanation. 

Answer: Upon your suggestion, we reorganized the article to comprehend the data and explanation.

Issue 21: Line 379: Do all the microorganisms under stress conditions make spores?

Answer: Microorganisms with strong stress resistance will make spores under thermophilic condition, including most fungi and molds, as well as bacteria like Bacillus.

Issue 22: Line 416 How does the decrease in fungal fragments corelates with the inactivation of miscellaneous bacteria?

Answer: The heating of the stack can realize the inactivation of miscellaneous bacteria [8], but only white rot fungi and brown rot fungi can metabolize normally under high temperature [9]. Therefore, with the temperature rising during the composting process, the fungi occupy the dominant position of the microbial community in the heap and simultaneously realize the extinguishment of the miscellaneous bacteria.

Issue 23: The names of the organisms should be italicized in the text and the references. 

Answer: Upon your suggestion, all the names of the organisms have been revised to be italicized in the text and the references.

Reference:

[1] Qiao C.C., Penton C.R., Liu C. et al., 2021. Patterns of fungal community succession triggered by C/N ratios during composting. J. Hazard. Mater. 401: 123344.

[2] Zheng X., Wang J., Zhang C.B., et al. 2022. Influence of dissolved organic matter on methylmercury transformation during aerobic composting of municipal sewage sludge under different C/N ratios. J. Environ. Sc. 119: 130-138.

[3] He Z.H., Lin H., Hao J.W., et al. 2018. Impact of vermiculite on ammonia emissions and organic matter decomposition of food waste during composting. Bioresour. Technol. 263: 548-554.

[4] Jiang Z.W., Meng Q.R., Niu Q.Q. et al. 2020. Understanding the key regulatory functions of red mud in cellulose breakdown and succession of β-glucosidase microbial community during composting. Bioresour. Technol. 318: 124265.

[5] Hestmark K.V., Fernandez-Bayo J.D., Harrold D.R., et al. 2019. Compost induces the accumulation of biopesticidal organic acids during soil biosolarization. Resour. Conserv. Recy. 143: 27-35.

[6] Zhang X., Ma D.C., Lv J.H., et al. 2022. Food waste composting based on patented compost bins: Carbon dioxide and nitrous oxide emissions and the denitrifying community analysis. Bioresour. Technol. 346: 126643.

[7] Syazni Z.K., Mitsuhiko K., Fadhil S., et al. 2022. Effect of enzymatic pre-treatment on thermophilic composting of shrimp pond sludge to improve ammonia recovery. Environ. Res. 204: 112299.

[8] Zhong X.Z., Sun Z.Y., Wang S.P., et al. 2020. Minimizing ammonia emissions from dairy manure composting by biofiltration using a pre-composted material as the packing media. Waste Manage. 102 (1): 569-578.

[9] Zhang C.S., Xu Y., Zhao M.H., Rong H.W., Zhang K.F., 2018. Influence of inoculating white-rot fungi on organic matter transformations and mobility of heavy metals in sewage sludge based composting. J. Hazard. Mater. 344 (15): 163-168.

Round 2

Reviewer 2 Report

Please see below the comments for the manuscript. 

1. The title mentions "An Alternative to Vermiculite". However, there is no comparison of the results obtained to the mentioned compound. How can the coral sand be labelled as an alternative if no comparison is made. And why alternative to Vermiculite only and not to other compounds with similar functionality. 
2. low pH means acidic, instead of writing weak acidic the authors should just mention the pH value that will give the readers an estimation of the process. 
3. The resolution of the graphs must be improved. The pixels of the figures are not well adjusted. The authors should use a minimum of 300 dpi to ensure the pixels in the figures do not zoom out. If a software is being used to generate the figures then use the print option for the figures.  Figure 2 avoid using the abbreviations as the readers might get confused or provide legends for the abbreviations. Avoid overuse of abbreviations, it shifts the focus from reading and understanding the manuscript to remembering and looking for the abbreviations. 
4. Line 142 "organic substances such as cellulose and lignin in the substrate did not de-142 compose completely without forced ventilation" The authors are only measuring the total carbon, how it was deduced that the carbon is coming from cellulose and lignin and not from hemicellulose, polymers or other carbon sources.  
5. Line 145 (version 2) is it a heresy or a fact. No reference to support the claim. The authors should provide reference for all the claims mentioned or omit from the manuscript where relevant source of information or reference is not available. 
6. The observations made and inferences deduced from the experimental work is compared to the prior art to ascertain that the hypothesis made are well supported by data. In the absence of such practice the results are either exceptional or mere speculations of "X is happening due to change in Y". The results are missing adequate support from the prior work to strengthen the results published from the observations made during the experimental work. For example: Section 3.1 No support of the results obtained by the previous done work. The authors should mention incase the observations are novel and never have been observed in the work done prior by the researchers in the field. Similarly the results are not well supported in the other sections also. 
6. Line 324 "According to the bioorganic fertilizer standard of the China Ministry of Agriculture", Line 404 "Among the known compost-403 ing microorganisms, white rot fungi and brown rot fungi can tolerate high temperatures" Heresy or fact. No references.

Author Response

We would like to express our sincere gratitude again to you and the reviewers for your kind help and valuable suggestions on our manuscript entitled “ An alternative to vermiculite: composting on tropical islands using coral sand to enhance nitrogen retention during ventilation ”. Upon your request, we carefully revised the manuscript and all the revisions were marked in yellow color in the revised manuscript, and we have also prepared this cover letter which contains the following materials. (1) A detailed response to reviewers’ comments. (2) A list of major changes in the revised manuscript.
